# A graph-theoretic approach to multitasking

**Noga Alon**[*]
Tel-Aviv University

**Daniel Reichman**[†]
UC Berkeley

**Igor Shinkar**[*]
UC Berkeley

**Tal Wagner**[*]
MIT

**Sebastian Musslick**
Princeton University

**Jonathan D. Cohen** [‡]
Princeton University

**Thomas L. Griffiths**
UC Berkeley

**Biswadip Dey**
Princeton University

**Kayhan Ozcimder**
Princeton University

## Abstract

A key feature of neural network architectures is their ability to support the simultaneous interaction among large numbers of units in the learning and processing of representations. However, how the richness of such interactions trades off against the ability of a network to simultaneously carry out multiple independent processes – a salient limitation in many domains of human cognition – remains largely unexplored. In this paper we use a graph-theoretic analysis of network architecture to address this question, where tasks are represented as edges in a bipartite graph $G = (A \cup B, E)$. We define a new measure of multitasking capacity of such networks, based on the assumptions that tasks that *need* to be multitasked rely on independent resources, i.e., form a matching, and that tasks *can* be multitasked without interference if they form an induced matching. Our main result is an inherent tradeoff between the multitasking capacity and the average degree of the network that holds *regardless of the network architecture*. These results are also extended to networks of depth greater than 2. On the positive side, we demonstrate that networks that are random-like (e.g., locally sparse) can have desirable multitasking properties. Our results shed light into the parallel-processing limitations of neural systems and provide insights that may be useful for the analysis and design of parallel architectures.

## 1 Introduction

One of the primary features of neural network architectures is their ability to support parallel distributed processing [RMG+86]. The decentralized nature of biological and artificial nets results in greater robustness and fault tolerance when compared to serial architectures such as Turing machines. On the other hand, the lack of a central coordination mechanism in neural networks can result in interference between units (neurons) and such interference effects have been demonstrated in several settings such as the analysis of associative memories [AGS85] and multitask learning [MC89].

---

[*]Equal contribution.

[†]Equal contribution. Supported by DARPA contract N66001-15-2-4048, Value Alignment in Autonomous Systems and Grant: 2014-1600, Sponsor: William and Flora Hewlett Foundation, Project Title: Cybersecurity and Internet Policy

[‡]This publication was made possible through the support of a grant from the John Templeton Foundation. The opinions expressed in this publication are those of the authors and do not necessarily reflect the views of the John Templeton Foundation

Understating the source of such interference and how it can be prevented has been a major focus of recent research (see, e.g., [KPR$^+$17] and the references therein).

While the stark limitation of our ability to carry out multiple tasks simultaneously, i.e., *multitask*, is one of the most widely documented phenomena in cognitive psychology [SS77], the sources for this limitation are still unclear. Recently, a graph-theoretic model [FSGC14, MDO$^+$16] has suggested that interference effects may explain the limitations of the human cognitive system in performing multiple task processes at the same time. This model consists of a simple 2-layer feed-forward network represented by a bipartite graph $G = (A \cup B, E)$ wherein the vertex set is partitioned into two disjoint sets of nodes $A$ and $B$, representing the inputs and the outputs of tasks respectively. An edge $(a, b) \in E$ corresponds to a directed pathway from the input layer to the output layer in the network that is taken to represent a cognitive process (or task[4]) that maps an input to an output. In more abstract terms, every vertex in $a \in A$ is associated with a set of inputs $I_a$, every vertex in $B$ is associated with a set of outputs $O_b$ and the edge $(a, b)$ is associated with a function $f_{a,b} : I_a \to O_b$.
[5] In this work, we also consider deeper architectures with $r > 2$ layers, where edges correspond to mappings between nodes from consecutive layers and a path $P$ from the input (first) layer to the output (last) layer is simply the composition of the mappings on the edges in $P$. The model above is quite general and simple modifications of it may apply to other settings. For example, we can assume the vertices in $A$ are senders and vertices in $B$ are receivers and that a task associated with an edge $e = (a, b)$ is transmitting information from $a$ to $b$ along a communication channel $e$.

Given a 2-layer network, a *task set* is a set of edges $T \subseteq E$. A key assumption made in [MDO$^+$16, FSGC14] that we adopt as well is that all task sets that need to be multitasked in parallel form a *matching*, namely, no two edges in $T$ share a vertex as an endpoint. This assumption reflects a limitation on the parallelism of the network that is similar to the Exclusive Read Exclusive Write (EREW) model in parallel RAM, where tasks cannot simultaneously read from the same input or write to the same output. Similarly, for depth $r > 2$ networks, task sets correspond to *node disjoint* paths from the input layer to the output layer. For simplicity, we shall mostly focus from now on the depth 2 case with $|A| = |B| = n$.

In [MDO$^+$16, FSGC14] it is suggested that concurrently executing two tasks associated with two (disjoint) edges $e$ and $f$ will result in interference if $e$ and $f$ are connected by a third edge $h$. The rationale for this interference assumption stems from the distributed operation of the network that may result in the task associated with $h$ becoming activated automatically once its input and output are operating, resulting in interference with the tasks associated with $e$ and $f$. Therefore, [MDO$^+$16, FSGC14] postulate that all tasks within a task set $T$ can be performed in parallel without interferences only if the edges in $T$ form an *induced* matching. Namely, no two edges in $T$ are connected by a third edge. Interestingly, the induced matching condition also arises in the communication setting [BLM93, AMS12, CK85], where it is assumed that messages between senders and receivers can be reliably transmitted if the edges connecting them forms an induced matching. Following the aforementioned interference model, [MDO$^+$16] define the multitasking capability of a bipartite network $G$ as the maximum cardinality of an induced matching in $G$.

It has been demonstrated that neural network architectures are subject to a fundamental tradeoff between learning efficiency that is promoted by an economic use of shared representations between tasks, on the one hand, and the ability of to execute multiple tasks independently, on the other hand [MSÖ$^+$17]. Namely, it is suggested that as the average degree $d$ ("efficiency of representations" – larger degree corresponds to more economical use of shared representations between tasks) of $G$ increases, the "multitasking ability" should decay in $d$ [FSGC14]. That is, the cardinality of the maximal induced matching should be upper bounded by $f(d)n$ with $\lim_{d \to \infty} f(d) = 0$. This prediction was tested and supported on certain architectures by numerical simulations in [MDO$^+$16, FSGC14], where it was suggested that environmental constraints push towards efficient use of representations which inevitably limits multitasking. Establishing such as a tradeoff is of interest, as

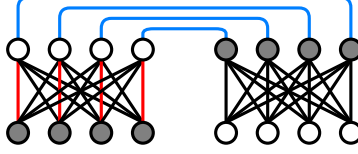

Figure 1: In the depicted bipartite graph, the node shading represents the bipartition. The blue edges form an induced matching, which represents a large set of tasks that can be multitasked. However, the red edges form a matching in which the largest induced matching has size only 1. This represents a set of tasks that greatly interfere with each other.

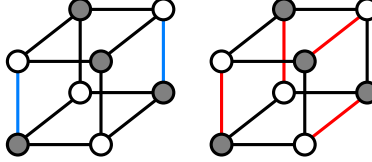

Figure 2: Hypercube on $8$ nodes. Node shading represents the bipartition. On the left, the blue edges form an induced matching of size $2$. On the right, the red edges form a matching of size $4$ whose largest induced matching has size 1. Hence the multitasking capacity of the hypercube is at most $1/4$.

it can identify limitations of artificial nets that rely on shared representations and aid in designing systems that attain an optimal tradeoff. More generally, establishing a connection between graph-theoretic parameters and connectionist models of cognition consists of a new conceptual development that may apply to domains beyond multitasking.

Identifying the multitasking capacity of $G = (A \cup B, E)$ with the size of its maximal induced matching has two drawbacks. First, the existence of some, possibly large, set of tasks that can be multitasked does not preclude the existence of a (possibly small) set of critical tasks that greatly interfere with each other (e.g., consider the case in which a complete bipartite graph $K_{d,d}$ occurs as a subgraph of $G$. This is illustrated in Figure 1). Second, it is easy to give examples of graphs (where $|A| = |B| = n$) with average degree $\Omega(n)$ that contain an induced matching of size $n/2$ (for example, two copies of complete bipartite graph connected by a matching: see Figure 1 for an illustration). Hence, it is impossible to upper bound the multitasking capacity of every network with average degree $d$ by $f(d)n$ with $f$ vanishing as the average degree $d$ tends infinity. Therefore, the generality of the suggested tradeoff between efficiency and concurrency is not clear under this definition.

Our main contribution is a novel measure of the multitasking capacity that is aimed at solving the first problem, namely networks with "high" capacity which contain a "small" task set whose edges badly interfere with one another. In particular, for a parameter $k$ we consider *every* matching of size $k$, and ask whether every matching $M$ of size $k$ contains a large *induced* matching $M' \subseteq M$. This motivates the following definition (see Figure 2 for an illustration).

**Definition 1.1.** *Let $G = (A \cup B, E)$ be a bipartite graph with $|A| = |B| = n$, and let $k \in \mathbb{N}$ be a parameter. We say that $G$ is a $(k, \alpha(k))$-multitasker if for every matching $M$ in $G$ of size $|M| \leq k$, there exists an induced matching $M' \subseteq M$ such that*

$$|M'| \geq \alpha(k)|M|.$$

*We will say that a graph $G$ is an $\alpha$-multitasker if it is $(n, \alpha)$-multitasker.*

*The parameter $\alpha \in (0, 1]$ measures the multitasking capabilities of $G$, and the larger $\alpha$ is the better multitasker $G$ is considered. We call the parameter $\alpha(k) \in (0, 1]$ the multitasking capacity of $G$ for matchings of size $k$.*

Definition 1.1 generalizes to networks of depth $r > 2$, where instead of matchings, we consider first layer to last layer node disjoint paths, and instead of induced matchings we consider induced paths, i.e., a set of disjoint paths such that no two nodes belonging to different paths are adjacent.

The main question we shall consider here is what kind of tradeoffs one should expect between $\alpha, d$ and $k$. In particular, which network architectures give rise to good multitasking behavior? Should we

expect "multitasking vs. multiplexing": namely, $\alpha$ tending to zero with $d$ for all graphs of average degree $d$? While our definition of multitasking capacity is aimed at resolving the problem of small task sets that can be poorly multitasked, it turns out to be also related also to the "multitasking vs. multiplexing" phenomena. Furthermore, our graph-theoretic formalism also gives insights as to how network depth and interference are related.

## 1.1 Our results

We divide the presentation of the results into two parts. The first part discusses the case of $d$-regular graphs, and the second part discusses general graphs.

**The $d$-regular case:** Let $G = (A \cup B, E)$ be a bipartite $d$-regular graph with $n$ vertices on each side. Considering the case of $k = n$, i.e., maximal possible induced matchings that are contained in a *perfect matching* (that is a matching of cardinality $n$), we show that if a $d$-regular graph is an $(n, \alpha(n))$-multitasker, then $\alpha(n) = O(1/\sqrt{d})$. Our upper bound on $\alpha(n)$ establishes an inherent limitation on the multitasking capacity of any network. That is, for any infinite family of networks with average degree tending to infinity it holds that $\alpha(n)$ must tend to 0 as the degree grows. In fact, we prove that degree of the graph $d$ constrains the multitasking capacity also for task sets of smaller sizes. Specifically, for all $k$ that is sufficiently larger than $\Omega(n/d)$ it holds that $\alpha(k)$ tends to 0 as $d$ increases. In this version of the paper we prove this result for $k > n/d^{1/4}$. The full version of this paper [ACD$^+$] contains the statement and the result that holds for all $d > \Omega(\frac{n}{d})$.

**Theorem 1.2.** *Let $G = (A \cup B, E)$, be a d-regular $(k, \alpha(k))$-multitasker graph with $|A| = |B| = n$. If $n/d^{1/4} \leq k \leq n$, then $\alpha(k) \leq O(\frac{n}{k\sqrt{d}})$. In particular, there exists a perfect matching in G that does not contain an induced matching of size larger than $O(n/\sqrt{d})$.*

For task sets of size $n$, Theorem 1.2 is tight up to logarithmic factors, as we provide a construction of an infinite family of $d$-regular graph, where every matching of size $n$ contains an induced matching of size $\Omega(\frac{1}{\sqrt{d \log d}})$. The precise statement appear in the full version of the paper [ACD$^+$].

For arbitrary values of $k \leq n$ it is not hard to see that every $d$-regular graph achieves $\alpha(k) \geq \frac{1}{2d}$. We show that this naive bound can be asymptotically improved upon, by constructing an $\alpha$-multitaskers with $\alpha = \Omega(\frac{\log d}{d})$. The construction is based on bipartite graphs which have good spectral expansion properties. For more details see the full version of the paper [ACD$^+$].

We also consider networks of depth $r > 2$ [6]. We generalize our ideas for depth 2 networks by upper-bounding the multitasking capacity of arbitrary $d$-regular networks of depth $r$ by $O((r/d \ln(r))^{1-1/r})$. Observe that as we show that there are $d$-regular bipartite graphs with $\alpha(n) = \frac{1}{\sqrt{d \log d}}$, this implies that for tasks sets of size $n$, networks of depth $2 < r \ll d$ incur interference which is strictly worse than depth 2 networks. We believe that interference worsens as $r$ increases to $r + 1$ (for $r > 2$), although whether this is indeed the case is an open question.

**The irregular case:** Next we turn to arbitrary, not necessarily regular, graphs. We show that for an arbitrary bipartite graph with $n$ vertices on each side and *average* degree $d$ its multitasking capacity $\alpha(n)$ is upper bounded by $O\left(\frac{\log n}{d}\right)^{1/3}$. That is, when the average degree is concerned, the multitasking capacity of a graph tends to zero, provided that the average degree of a graph is $\omega(\log n)$.

**Theorem 1.3.** *Let $G = (A \cup B, E)$, be a bipartite graph of average degree $d$ with $|A| = |B| = n$. If G is an $\alpha$-multitasker then $\alpha \leq O((\frac{\log n}{d})^{1/3})$.*

For dense graphs satisfying $d = \Omega(n)$ (which is studied in [FSGC14]), we prove a stronger upper bound of $\alpha(n) = O(\frac{1}{\sqrt{n}})$ using the Szemerédi regularity lemma. See Theorem 3.9 for details.

We also show that there are multitaskers of average degree $\Omega(\log \log n)$, with $\alpha > 1/3 - \epsilon$. Hence, in contrast to the regular case, for the multitasking capacity to decay with *average* degree $d$, we must assume that $d$ grows faster than $\log \log n$. The details behind this construction, which build on ideas in [Pyb85, PRS95], appear in full version of this paper [ACD$^+$].

Finally, for any $d \in \mathbb{N}$ and for all $\alpha \in (0, 1/5)$ we show a construction of a graph with average degree $d$ that is a $(k, \alpha)$-multitaskers for all $k \leq \Omega(n/d^{1+4\alpha})$. Comparing this to the foregoing results, here we do not require that $d = O(\log \log n)$. That is, allowing larger values of $d$ allows us to construct networks with constant multitasking capacities, albeit only with respect to matchings whose size is at most $n/d^{1+4\alpha}$. See Theorem 3.10 for details.

## 2    Preliminaries

A matching $M$ in a graph $G$ is a set of edges $\{e_1, ..., e_m\}$ such that no two edges in $M$ share a common vertex. If $G$ has $2n$ vertices and $|M| = n$, we say that $M$ is a perfect matching. By Hall Theorem, every $d$-regular graph with bipartition $(A, B)$ has a perfect matching. A matching $M$ is *induced* if there are no two distinct edges $e_1, e_2$ in $M$, such that there is an edge connecting $e_1$ to $e_2$. Given a graph $G = (V, E)$ and two disjoint sets $A, B \subseteq V$ we let $e(A, B)$ be the set of edges with one endpoint in $A$ and the other in $B$. For a subset $A$, $e(A)$ is the set of all edges contained in $A$. Given an edge $e \in E$, we define the graph $G/e$ obtained by contracting $e = (u, v)$ as the graph with a vertex set $(V \cup v_e) \setminus \{u, v\}$. The vertex $v_e$ is connected to all vertices in $G$ neighboring $u$ or $v$. For all other vertices $x, y \in V \setminus \{u, v\}$, they form an edge in $G/e$ if and only if they were connected in $G$. Contracting a set of edges, and in particular contracting a matching, means contracting the edges one by one in an arbitrary order.

Given a subset of vertices $U \subseteq V$, the subgraph induced by $U$, denoted by $G[U]$ is the graph whose vertex set is $U$ and two vertices in $U$ are connected if and only if they are connected in $G$. For a set of edges $E' \subseteq E$, denote by $G[E']$ the graph induced by all vertices incident to an edge in $E'$. We will use the following simple observation throughout the paper.

**Lemma 2.1.** *Let $M$ be a matching in $G$, and let $d_{avg}$ be the average degree of $G[M]$. If we contract all edges in $M$ in $G[M]$, then the resulting graph $\widetilde{G}[M]$ has average degree at most $2d_{avg} - 2$.*

*Proof.* $G[M]$ contains $2|M|$ vertices and $d_{avg}|M|$ edges. The result follows as $\widetilde{G}[M]$ has $|M|$ vertices and at most $d_{avg}|M| - |M|$ edges. $\square$

An *independent set* in a graph $G = (V, E)$ is a set of vertices that do not span an edge. We will use the following well known fact attributed to Turan.

**Lemma 2.2.** *Every $n$-vertex graph with average degree $d_{avg}$ contains an independent set of size at least $\frac{n}{d_{avg}+1}$.*

Let $G = (V, E)$ be a bipartite graph, $k$ an integer and $\alpha \in (0, 1]$, a parameter. We define the $(\alpha, k)$-matching graph $H(G, \alpha, k) = (L, R, F)$ to be a bipartite graph, where $L$ is the set of all matchings of size $k$ in $G$, $R$ is the set of all induced matchings of size $\alpha k$ in $G$, and a vertex $v_M \in L$ (corresponding to matching $M$ of size $k$) is connected to a vertex $u_{M'}$ (corresponding to an induced matching $M'$ of size $\alpha k$) if and only if $M' \subseteq M$. We omit $\alpha, k, G$ from the notation of $H$ when it will be clear from the context. We will repeatedly use the following lemma in upper bounding the multitasking capacity in graph families.

**Lemma 2.3.** *Suppose that the average degree of the vertices in $L$ in the graph $H(G, \alpha, k)$ is strictly smaller than 1. Then $\alpha(k) < \alpha$.*

*Proof.* By the assumption, $L$ has a vertex of degree 0. Hence there exist a matching of size $k$ in $G$ that does not contain an induced matching of size $\alpha k$. $\square$

## 3    Upper bounds on the multitasking capacity

### 3.1    The regular case

In this section we prove Theorem 1.2 that upper bounds the multitasking capacity of arbitrary $d$-regular multitaskers. We start the proof of Theorem 1.2 with the case $k = n$. The following theorem shows that $d$-regular $(k = n, \alpha)$-multitaskers must have $\alpha = O(1/\sqrt{d})$.

**Theorem 3.1.** *Let $G = (A \cup B, E)$ be a bipartite $d$-regular graph where $|A| = |B| = n$. Then $G$ contains a perfect matching $M$ such that every induced matching $M' \subseteq M$ has size at most $\frac{9n}{\sqrt{d}}$.*

For the proof, we need bounds on the number of perfect matchings in $d$-regular bipartite graphs.

**Lemma 3.2.** *Let $G = (A, B, E)$, be a bipartite $d$-regular graph where $|A| = |B| = n$. Denote by $M(G)$ the number of perfect matchings in $G$. Then*

$$\left(\frac{d}{e}\right)^n \leq \left(\frac{(d-1)^{d-1}}{d^{d-2}}\right)^n \leq M(G) \leq (d!)^{n/d}.$$

The lower bound on $M(G)$ is due to Schrijver [Sch98]. The upper bound on $M(G)$ is known as Minc's conjecture, which has been proven by Bregman [Bre73].

*Proof of Theorem 3.1.* Consider $H(G, \alpha, n)$, where $\alpha$ will be determined later. Clearly $|R| \leq \binom{n}{\alpha n}^2 \leq \left(\frac{e}{\alpha}\right)^{2\alpha n}$. By the upper bound in Lemma 3.2, every induced matching of size $\alpha n$ can be contained in at most $(d!)^{(1-\alpha)n/d}$ perfect matchings. By the lower bound in Lemma 3.2, $|L| \geq \left(\frac{d}{e}\right)^n$. Therefore, the average degree of the the vertices in $L$ is at most

$$\frac{\left(\frac{e}{\alpha}\right)^{2\alpha n} \cdot (d!)^{(1-\alpha)n/d}}{\left(\frac{d}{e}\right)^n} \leq \frac{\left(\frac{e}{\alpha}\right)^{2\alpha n} \cdot (\sqrt{2\pi d}(\frac{d}{e})^d)^{(1-\alpha)n/d}}{\left(\frac{d}{e}\right)^n} = \left(\frac{e^3}{\alpha^2 d} \cdot (2\pi d)^{\frac{1-\alpha}{2\alpha d}}\right)^{\alpha n}.$$

Setting $\alpha > 2\sqrt{\frac{e^3}{d}}$ yields $\frac{e^3}{\alpha^2 d} < \frac{1}{2}$, and it can be verified that $(2\pi d)^{\frac{1-\alpha}{2\alpha d}} < 2$ for all such $\alpha$. Therefore in this setting, the average degree of the vertices in $L$ is smaller than 1, which concludes the proof by Lemma 2.3. This completes the proof of the theorem. $\qquad\square$

We record the following simple observation, which is immediate from the definition.

**Proposition 3.3.** *If $G$ is a $(k, \alpha)$-multitasker, then for all $1 < \beta \leq n/k$, the graph $G$ is a $(\beta k, \frac{\alpha}{\beta})$-multitasker.*

Theorem 1.2 follows by combining Theorem 3.1 with (the contrapositive of) Proposition 3.3.

## 3.2   Upper bounds for networks of depth larger than 2

A graph $G = (V, E)$ is a *network* with $r$ layers of width $n$ and degree $d$, if $V$ is partitioned into $r$ independent sets $V_1, \ldots, V_r$ of size $n$ each, such that each $(V_i, V_{i+1})$ induces a $d$-regular bipartite graph for all $i < r$, and there are no additional edges in $G$.

A *top-bottom* path in $G$ is a path $v_1, \ldots, v_r$ such that $v_i \in V_i$ for all $i \leq r$, and $v_i, v_{i+1}$ are neighbors for all $i < r$.

A set of node-disjoint top-bottom paths $p_1, \ldots, p_k$ is called *induced* if for every two edges $e \in p_i$ and $e' \in p_j$ such that $i \neq j$, there is no edge in $G$ connecting $e$ and $e'$.

**Fact 3.4.** *A set of node-disjoint top-bottom paths $p_1, \ldots, p_k$ is induced if and only if for every $i < r$ it holds that $(p_1 \cup \ldots \cup p_k) \cap E(V_i, V_{i+1})$ is an induced matching in $G$.*

We say that a network $G$ as above is a $(k, \alpha)$-multitasker if every set of $k$ node-disjoint top-bottom paths contains an induced subset of size at least $\alpha k$.

**Theorem 3.5.** *If $G$ is an $(n, \alpha)$-multitasker then $\alpha < e \left(\frac{e \cdot r}{d \ln(r)}\right)^{1-\frac{1}{r}}$.*

*Proof.* Let $H = (L, R; E_H)$ be the bipartite graph in which side $L$ has a node for each set of $n$ node-disjoint top-bottom paths in $G$, side $R$ has a node for each induced set of $\alpha n$ node-disjoint top-bottom paths in $G$, and $P \in L$, $P' \in R$ are adjacent iff $P' \subset P$. Let $D$ be the maximum degree of side $R$. We wish to upper-bound the average degree of side $L$, which is upper-bounded by $D|R|/|L|$.

$|R|$ is clearly upper bounded by $\binom{n}{\alpha n}^r$. It is a simple observation that $|L|$ equals $\prod_{i<r} m_i$, where $m_i$ denotes the number of perfect matchings in the bipartite graph $G[V_i \cup V_{i+1}]$. Since this graph is

$d$-regular, by the Falikman-Egorichev proof of the Van der Waerden conjecture ([Fal81], [Ego81]), or by Schrijver's lower bound, we have $m_i \geq (d/e)^n$ and hence $|L| \geq (d/e)^{n(r-1)}$. To upper bound $D$, fix $P' \in R$, and let $G'$ be the network resulting by removing all nodes and edges in $P'$ from $G$. This removes exactly $\alpha n$ nodes from each layer $V_i$; denote by $V_i'$ the remaining nodes in this layer in $G'$. It is a straightforward observation that $D$ equals the number of sets of $(1-\alpha)n$ node-disjoint top-bottom paths in $G'$. Each such set decomposes into $M_1, \ldots, M_{r-1}$ such that $M_i$ is a perfect matching on $G'[V_i', V_{i+1}']$ for each $i < r$. Therefore $D \leq \prod_{i-1} m_i'$ where $m_i'$ denotes the number of perfect matchings in $G'[V_i', V_{i+1}']$. The latter is a bipartite graph with $(1-\alpha)n$ nodes on each side and maximum degree $d$, and hence by the Bregman-Minc inequality, $m_i' \leq (d!)^{(1-\alpha)n/d}$. Consequently, $D \leq (d!)^{(1-\alpha)n(r-1)/d}$.

Putting everything together, we find that the average degree of side $L$ is upper bounded by

$$\frac{D|R|}{|L|} \leq \frac{(d!)^{(1-\alpha)n(r-1)/d} \cdot \binom{n}{\alpha n}^r}{(d/e)^{n(r-1)}} \leq \frac{(\sqrt{2\pi d}(d/e)^d)^{(1-\alpha)n(r-1)/d} \cdot (\frac{e}{\alpha})^{\alpha nr}}{(d/e)^{n(r-1)}}$$

$$= \left( (2\pi d)^{\frac{1-\alpha}{2\alpha d}} \cdot \frac{e}{d} \left(\frac{e}{\alpha}\right)^{\frac{r}{r-1}} \right)^{\alpha n(r-1)}. \tag{1}$$

For $C = r/\ln(r)$ we will show that if $\alpha \geq e(eC/d)^{1-\frac{1}{r}}$ then above bound is less than $1$, which implies side $L$ has a node of degree $0$, a contradiction. To this end, note that for this $\alpha$ we have

$$\frac{e}{d} \left(\frac{e}{\alpha}\right)^{\frac{r}{r-1}} \leq \frac{1}{C} = \frac{\ln(r)}{r}, \tag{2}$$

and

$$(2\pi d)^{(1-\alpha)/(2\alpha d)} \leq (2\pi d)^{1/(2\alpha d)} \leq (2\pi d)^{1/(2eC^{1-1/r}d^{1/r})}.$$

**Fact 3.6.** *For every constants $\gamma, \beta > 0$, the function $f(d) = (\gamma d)^{1/(\beta d^{1/r})}$ is maximized at $d = e^r/\gamma$, and $f(e^r/\gamma) = e^{r\gamma^{1/r}/\beta e}$.*

Plugging this above (and using $r \geq 2$), we obtain

$$(2\pi d)^{(1-\alpha)/(2\alpha d)} \leq (2\pi d)^{1/(2eC^{1-1/r}d^{1/r})} \leq e^{r(2\pi eC)^{1/r}/(2Ce^2)} \leq e^{\ln(r)\sqrt{2\pi} \cdot r^{1/r}/(2e^{3/2})} \leq \sqrt{r},$$

and plugging this with Equation (2) into Equation (1) yields $\frac{D|R|}{|L|} < 1$, as required. $\qquad\square$

### 3.3 The irregular case

Below we consider general (not necessarily regular) graphs with average degree $d$, and prove Theorem 1.3. In order to prove it, we first show a limitation on the multitasking capacity of graphs where the average degree of a graph is $d$, and the maximum degree is bounded by a parameter $\Delta$.

**Theorem 3.7.** *Let $G$ be a bipartite graph with $n$ nodes on each side, average degree $d$, and maximum degree $\Delta$. If $G$ is an $\alpha$-multitasker, then $\alpha < O(\Delta^{\frac{1}{3}}/d^{\frac{2}{3}})$.*

A proof of Theorem 3.7 can be found in the full version of this paper [ACD$^+$].

Note that Theorem 3.7 does not provide any nontrivial bounds on $\alpha$ when $\Delta$ exceeds $d^2$. However, we use it to prove Theorem 1.3, which establishes nearly the same upper bound with no assumption on $\Delta$. To do so we need the following lemma, which is also proved in the full version of this paper [ACD$^+$].

**Lemma 3.8.** *Every bipartite graph with $2n$ vertices and average degree $d > 4\log n$ contains a subgraph in which the average degree is at least $b = \frac{d}{4\log n}$ and the maximum degree is at most $2b$.*

We can now prove Theorem 1.3.

*Proof of Theorem 1.3.* By Lemma 3.8 $G$ contains a subgraph with average degree $b \geq d/(4\log n)$ and maximum degree at most $2b$. The result thus follows from Theorem 3.7. $\qquad\square$

As in the regular case, for smaller values of $k$ we can obtain a bound of $\alpha = O(\sqrt{\frac{n}{dk}})$ for $(k, \alpha)$-multitaskers. See the full version of this paper [ACD$^+$] for the precise details.

When the graph is dense, we prove the following better upper bounds on $\alpha$.

**Theorem 3.9.** *Let $G$ be a bipartite graph with $n$ vertices on each side, and average degree $d = \Omega(n)$. If $G$ is an $\alpha$-multitasker, then $\alpha < O((\frac{1}{n})^{1/2})$.*

*Proof.* By the result in [PRS95] (see Theorem 3) the graph $G$ contains a $d'$-regular bipartite graph with $d' = \Omega(n)$. The result thus follows from our upper bound for regular graphs as stated in Theorem 1.2. □

### 3.4 A simple construction of a good multitasker

We show that for small constants $\alpha$, we may achieve a significant increase in $k$ show existence of a $(O(n/d^{1+4\alpha}), \alpha)$-multitaskers for any $0 < \alpha < 1/5$.

**Theorem 3.10.** *Fix $d \in \mathbb{N}$, and let $n \in \mathbb{N}$ be sufficiently large. For a fixed $0 < \alpha < 1/5$, there exists a $(k, \alpha)$-multitasker with $n$ vertices on each side, average degree $d$, for all $k \leq \Omega(n/d^{1+4\alpha})$.*

*Proof.* It is known (see, e.g., [FW16]) that for sufficiently large $n$, there exist an $n$-vertex graph $G = (V, E)$ with average degree $d$ such that every subgraph of $G$ of size $s \leq O(n/d^{1+4\alpha})$ has average degree at most $\frac{1}{2}(\frac{1}{\alpha} - 1)$. Define a bipartite graph $H = (A \cup B, E_H)$ such that $A$ and $B$ are two copies of $V$, and for $a \in A$ and $b \in B$ we have $(a, b) \in E_H$ if and only if $(a, b) \in E$. We get that the average degree of $H$ is $d$, and for any two $A' \subseteq A$ and $B' \subseteq B$ such that $|A'| = |B'| \leq s/2$, the average degree of $H[A' \cup B']$ is at most $\frac{1}{\alpha} - 1$. Consider a matching $M$ of size $s/2$ in $H$. By Lemma 2.1, if we contract all edges of the matching, we get a graph of average degree at most $\frac{2}{\alpha} - 1$. By Lemma 2.2, such a graph contains an independent set of size at least $\frac{1}{2}\alpha|M|$, which corresponds to a large induced matching contained in $M$. This concludes the proof of the theorem. □

## 4 Conclusions

We have considered a new multitasking measure for parallel architectures that is aimed at providing quantitative measures of parallel processing capabilities of neural systems. We established an inherent tradeoff between the density of the network and its multitasking capacity that holds for every graph that is sufficiently dense. This tradeoff is rather general and it applies to regular graphs, to irregular graphs and to layered networks of depth greater than 2. We have also obtained quantitative insights. For example, we have provided evidence that interference increases as depth increases from 2 to $r > 2$, and demonstrated that irregular graphs allow for better multitasking than regular graphs for certain edge densities. Our findings are also related to recent efforts in cognitive neuroscience to pinpoint the reason for the limitations people experience in multiasking control demanding tasks.

We have found that networks with pseudorandom properties (locally sparse, spectral expanders) have good multitasking capabilities. Interestingly, previous works have documented the benefits of random and pseudorandom architectures in deep learning, Hopfield networks and other settings [ABGM14, Val00, KP88]. Whether there is an underlying cause for these results remains an interesting direction for future research.

Our work is limited in several aspects. First, our model is graph-theoretic in nature, focusing exclusively on the adjacency structure of tasks and does not consider many parameters that emerge in biological and artificial parallel architectures. Second, we do not address tasks of different weights (assuming all tasks have the same weights), stochastic and probabilistic interference (we assume interference occurs with probability 1) and the exact implementation of the functions that compute the tasks represented by edges. A promising avenue for future work will be to evaluate the predictive validity of $\alpha$, that is, the ability to predict parallel processing performance of trained neural networks from corresponding measures of $\alpha$.

To summarize, the current work is directed towards laying the foundations for a deeper understanding of the factors that affect the tension between efficiency of representation, and flexibility of processing in neural network architectures. We hope that this will help inspire a parallel proliferation of efforts to further explore this area.

## Footnotes

[4]We view a task as constituting a simple mechanistic instantiation of a cognitive process, consistent with Neisser's original definition [Nei67]. According to this definition a task process (e.g. color naming) is a mapping from an input space (e.g. colors) to an output space (verbal). Within this framework the decision of what constitutes an input space for a task is left to the designer and may be problem-specific. The modeling of more complex tasks might require to extend this framework to multidimensional input spaces. This would allow to capture scenarios in which tasks are partially overlapping in terms of their input and output spaces.

[5]The function $f_{a,b}$ is hypothesized to be implemented by a gate used in neural networks such as sigmoid or threshold gate.

[6]We think of $r$ as a constant independent of $n$ and $d$ as tending to infinity with $n$.

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
