[Reviews · NeurIPS 2017]

Reviewer 1



This paper solves an excellent theoretical problem: what are the limits of inducable matchings in bipartite graphs? This is a novel formulation in graph theory and the paper provides near tight scaling laws for the problem. Indeed, this problem is directly applicable in communication theory where interference and multi-user communication are accurately modeled by inducable matchings. 1. Why inducable matchings is a good model for multi-tasking is a bit unclear: in a multi-tasking problem what are the inputs and what are the outputs. It seems in a human or humanoid setting, the inputs are input sense streams and the outputs are action streams. In such a case, a task should use a subset of inputs to obtain a given output (for example, vision, hearing and tactile feedback to decide how to hold a paper cup). However, in the model, a task simply connects one input to one output. 2. The mathematical model of inducable matchings is quite interesting, and shows why the multi-tasking parameter could be of interest. The paper points out the pitfalls of using matchings as a proxy for multi-tasking and proposes this alternative formulation. Indeed, a future algorithmic problem is the calculation of some inducable matching coefficient given a graph. 3. The scaling laws connecting the degree of the graph to its multi-tasking ability is interesting, and it can potentially be used to design networks with appropriate overlaps for multi-tasking. While this model is not immediately applicable, the extremal graphs for inducable matchings can indeed inspire multitasking network architectures in future research. 4. Overall multi-tasking is an important problem and while the paper presents a fairly simplified model, the mathematical modeling has novelty and the results are quite interesting.

Reviewer 2



The paper considers an interesting ability of neural networks, named concurrent multitasking, and reveals its relation to the efficiency of shared representations, which is presented by the average degree d. This paper first defines the multitasking capacity as the maximal induced matching of every matching in the original graph of size from 1 to n. Then it proves that the multitasking capacity is upper bounded by f(d)n with f(d) approaches 0 as d grows. The paper shows upper bounds in both regular and irregular graphs, but with different orders of d. The topic studied in this paper is of interest. The result is meaningful and its proofs look ok. The result can give us some intuition about how to construct the neural network especially in the view of representation efficiency and multitasking capacity. This work will encourage more people to move on to the research about neural network in theory. There should be some intuitions or examples to explain why choosing d as the efficiency of representations and induced matching as the capacity of the multitasking. The proofs are somewhat technical and most of them is presented in the supplementary material, so I just check some of them.

Reviewer 3



The authors lay an ambitious groundwork for studying the multitasking capability of general neural network architectures. They prove a variety of theorems concerning behavior of their defined criteria--the multitasking capacity \alpha--and how it relates to graph connectivity and size via graph matchings and degree. Pros: The problem being studied is quite clearly motivated, and the authors present a genuinely novel new measure for attempting to understand the multitasking capability of networks. The authors fairly systematically calculate bounds on their parameter in a variety of different graph-theoretic scenarios. Cons: It's unclear how practically useful the parameter \alpha will be for real networks currently in use. Admittedly, this work is preliminary, and the authors point out this shortcoming in their conclusion. While it is likely the topic of an upcoming study, it would benefit the thrust of the paper considerably if some \alpha's were calculated for networks that are actually in use, or *some* sort of numerical comparison were demonstrated (ex: it's recently become popular to train neural networks to perform multiple tasks simultaneously--what does \alpha have to say about the topologies of these networks?) Plainly, it's not obvious from a first read of the paper that \alpha *means* anything short of being an abstract quantity one can calculate for graphs. Additional reads clarify the connection, but making some sort of empirical contact with networks used in practice (beyond the throwaway lines in the conclusion about pseudorandom graphs) would greatly clarify the practical use of (or even intuition for) the \alpha parameter. That said, this reviewer thinks the article should still be accepted in the absence of such an empirical study, because performing such a study would likely (at least) double the length of the paper (which would be silly). At the minimum, though, this reviewer would appreciate a bit more discussion of the practical use of the parameter.